# Current Advancement in Diagnosing Atrial Fibrillation by Utilizing Wearable Devices and Artificial Intelligence: A Review Study

**DOI:** 10.3390/diagnostics12030689

**Published:** 2022-03-11

**Authors:** Yu-Chiang Wang, Xiaobo Xu, Adrija Hajra, Samuel Apple, Amrin Kharawala, Gustavo Duarte, Wasla Liaqat, Yiwen Fu, Weijia Li, Yiyun Chen, Robert T. Faillace

**Affiliations:** 1Department of Medicine, New York City Health + Hospitals/Jacobi, Albert Einstein College of Medicine, The Bronx, New York, NY 10461, USA; xiaoboxumd@outlook.com (X.X.); adrija847@gmail.com (A.H.); apples@nychhc.org (S.A.); amrin.kharawala@outlook.com (A.K.); duarteg1@nychhc.org (G.D.); liaqatw@nychhc.org (W.L.); weijialisysu@gmail.com (W.L.); cheny36@nychhc.org (Y.C.); robert.faillace@nychhc.org (R.T.F.); 2Department of Medicine, Kaiser Permanente Santa Clara Medical Center, Santa Clara, CA 95051, USA; yf3255@gmail.com

**Keywords:** atrial fibrillation, artificial intelligence, wearable devices, machine learning

## Abstract

Atrial fibrillation (AF) is a common arrhythmia affecting 8–10% of the population older than 80 years old. The importance of early diagnosis of atrial fibrillation has been broadly recognized since arrhythmias significantly increase the risk of stroke, heart failure and tachycardia-induced cardiomyopathy with reduced cardiac function. However, the prevalence of atrial fibrillation is often underestimated due to the high frequency of clinically silent atrial fibrillation as well as paroxysmal atrial fibrillation, both of which are hard to catch by routine physical examination or 12-lead electrocardiogram (ECG). The development of wearable devices has provided a reliable way for healthcare providers to uncover undiagnosed atrial fibrillation in the population, especially those most at risk. Furthermore, with the advancement of artificial intelligence and machine learning, the technology is now able to utilize the database in assisting detection of arrhythmias from the data collected by the devices. In this review study, we compare the different wearable devices available on the market and review the current advancement in artificial intelligence in diagnosing atrial fibrillation. We believe that with the aid of the progressive development of technologies, the diagnosis of atrial fibrillation shall be made more effectively and accurately in the near future.

## 1. Background and Introduction

### 1.1. Atrial Fibrillation

Atrial fibrillation (AF) is defined as a type of supraventricular arrhythmia characterized by uncoordinated atrial activation which leads to ineffective atrial contraction [1]. It was first described as “auricular fibrillation” by William Harvey in 1628 and was thought to be the dissociation between the peripheral pulse and heartbeat [2]. Since then, AF has been studied extensively and has been broadly divided into different categories based on its nature. Paroxysmal AF is defined as AF that terminates spontaneously or with intervention within 7 days of onset. Persistent AF is continuous AF that lasts for more than 7 days [3]. Long-standing persistent AF is defined by uninterrupted AF for more than 12 months, while permanent AF is persistent AF with no rhythm control strategy pursued by the patient and physician.

The prevalence of AF increases with aging [4]. It is seen in 8–10% of people aged more than 80 years, while it occurs in less than 1% of the population aged 60 to 65 years [5]. Females have higher incidental rates compared to males [6]. It is also noted that people of European descent are more likely to have AF than African Americans [7]. Other risk factors are associated with the development of AF, including hypertension, obesity, diabetes, heart failure, ischemic heart disease, hyperthyroidism, chronic kidney disease, moderate to heavy alcohol use and smoking and sleep disordered breathing [8,9,10,11,12,13,14,15,16].

According to the Centers for Disease Control (CDC) of the United States database, AF contributes to approximately 158,000 deaths each year [17]. For more than two decades, the death rate from AF as a primary or contributing cause of death has been rising [18]. Each year, AF is the primary diagnosis in more than 454,000 hospitalizations in the US [19]. It is estimated that by 2030, about 12.1 million people will be diagnosed with AF in the United States [8]. Due to the large burden of this disease with increasing morbidities and mortalities, there have been significant advancements in terms of research for the treatment of AF. Apart from lifestyle modifications for risk reduction, treatment of AF can be categorized into three major categories: rate control, rhythm control and anticoagulation based on the CHA2DS2-VASc score and HAS-BLED score for stroke prevention [1,15,20]. Slowing the ventricular rate using AV nodal blocking agents such as beta blockers, non-dihydropyridine calcium channel blockers (verapamil and diltiazem) and digoxin improves the quality of life and decreases the risk of developing tachycardia-induced cardiomyopathy [1]. Alternatively, rhythm-control strategies using pharmacological cardioversion with class 3 anti-arrhythmic drugs based on Vaughan-Williams anti-arrhythmic drug classification, such as amiodarone, electric cardioversion with direct current or catheter ablation of AF foci in the atria, are used to restore and maintain normal sinus rhythm in patients with long-term AF [1,16,21]. In terms of anticoagulation, warfarin is used to prevent thromboembolism in patients with valvular AF, while new oral anticoagulants (NOACs) are often used in patients with non-valvular AF [1]. Recent advances in the treatment of AF have shown that left atrial appendage occlusion with the Watchman device is non-inferior to warfarin in preventing stroke in patients with non-valvular AF [22]. The medical field is evolving in the management of AF, yet due to the large burden of silent AF, diagnosing AF remains challenging. AF can be asymptomatic or present with symptoms such as palpitations, dyspnea, chest pain, decreased exercise tolerance and fatigue [14]. Many patients with silent AF may present with stroke as their first symptom [15]. The risk of cerebral embolism increases greatly in chronic AF, which has been estimated to account for approximately 50% of cardioembolic strokes. Atrial fibrillation is also associated with an increased risk of tachycardia-induced cardiomyopathy. Uncontrolled AF can eventually lead to reduced ventricular filling, increased left atrial pressures, hemodynamic instability, reduced cardiac output and morbidity [16].

### 1.2. Current Diagnosing Strategies and Challenges of AF

Statistics from the Global Burden of Disease study report that AF affects 2.5% to 3.2% of the population across many countries, with a worldwide prevalence of as high as 33.5 million cases [23]. The diagnostic challenges for AF are evident by the fact that in the USA alone, almost 700,000 people are estimated to have undiagnosed AF which contributes to a total of approximately three billion US healthcare expenses in a year [24,25]. On a 12-lead ECG, AF is characterized by irregular R-R intervals (when atrioventricular conduction is present) and the absence of P waves with irregular atrial activities [26]. In the SAFE trial, Hobbs et al. concluded that sufficient screening in both men and women after the age of 65 years old was cost-effective, leading to a significantly lower number of ischemic strokes with more diagnoses of AF [15]. This same strategy of using pulse palpation followed by ECG in individuals with an irregular pulse was recommended by atrial fibrillation guidelines in all patients older than 65 years [27]. However, studies have suggested that traditional screening tools utilizing ECG and pulse check are very likely to miss paroxysmal AF [28,29].

Research to address better diagnostic modalities is in progress. The mHealth Screening to Prevent Strokes (mSToPS) screening trial by Quer et al. found that the yield of ECG screening once or twice daily randomly for 30 s was low at 35% and 52%, respectively [28]. In addition, Yano et al. performed a post hoc analysis of daily ECG simulation from the TRENDS study and disclosed that daily snapshots of ECG monitoring could only detect 50% of patients who developed atrial tachycardia or AF as compared to cardiac implantable electronic devices [29]. This study showed that intermittent random monitoring of ECG, including 24 h ambulatory ECG recording, has low sensitivity (<20% for syncope and 35% for arrhythmia), especially in patients with a low pre-test likelihood of AF [28,29,30]. Other modalities include utilizing external loop recorders (ELRs) or triggered event recorders which can diagnose up to 60% of cases; however, these devices are usually bulkier and inconvenient to wear during daily activities [30]. Sejr et al. suggested that, in a comparison of readings for AF diagnosis (in patients known with stroke or TIA) on ELRs to cardiologist-verified AF on continuous ECG monitoring, the sensitivity of ELRs was 84% with a 98% specificity and a 68% positive predictive value. Thus, an automatic ELR may be considered to rule out AF, but it is not the best monitoring device for AF screening in patients after early stroke due to high false-positive values [31].

When implantable cardiac monitoring (ICM) was first introduced, the positive predictive value was low especially for short episodes of AF [32]. However, ICM technology has now improved tremendously with reduced size and continuous rhythm monitoring for up to three years, making it a cost-effective method of diagnosing arrhythmias [33,34]. Trials such as ASSERT-2 and REVEAL AF showed that the rate of detection of sub-clinical AF by ICM after 1 year was 34% and 27%, respectively [35,36]. These studies further demonstrated that AF is common in elderly individuals with cardiovascular risk factors [36,37]. In the CRYSTAL AF trial, investigators further proved that ECG monitoring with ICM was more sensitive for AF diagnosis after a cryptogenic stroke [38,39,40]. ICM allows a long-term surveillance and a higher detection rate of AF in high-risk patients. However, despite high sensitivity from other methods such as ELRs, ICM is frequently associated with a high number of false positives (up to 46% for AF detection) and a higher degree of artifact which reduces its specificity [32,41]. This paved the way for other monitoring devices such as pacemaker monitors and Holter monitors which have been proven to be more accurate in diagnosing AF [41]. In comparing the diagnostic efficiency of short-term Holter monitors to adhesive ECG patch monitors, significantly higher rates of arrhythmia detection were found on the 14-day patch as compared to the 24 h Holter monitor, which was further independently validated by multiple studies, including the EMBRACE trial [42,43,44]. Currently, these wearable devices and ECG patches for ambulatory ECG monitoring are the standard of care for the diagnosis of AF.

### 1.3. The Advent of Artificial Intelligence and Machine Learning in Atrial Fibrillation

The advancement in artificial intelligence technology has enabled its broad utilization in different medical fields including atrial fibrillation detection. With the rapid increase of digital clinical data, artificial intelligence, along with wearable devices, was able to extract and analyze different variables to detect and predict the groups of patients that were missed by using conventional detecting tools [45]. Machine learning algorithms can assist in processing electrophysiological data and images and are able to interpret and analyze large amounts of clinical data and possibly discover new traces and patterns.

In this study, we aim to present the latest review of wearable devices and the current advancement of artificial intelligence and machine learning in atrial fibrillation diagnosis and management [46,47].

## 2. Wearable Cardiac Monitoring Devices

### 2.1. Wearable Devices in Medicine

Wearable devices such as smartwatches, rings and wristbands are becoming increasingly popular (Table 1). These sophisticated sensors are gaining popularity as they help monitor health-related data such as heart rate, blood pressure and oxygen saturation [48]. Approximately 20% of US residents currently own a smart wearable device, and the global market of these devices is expected to reach USD 70 billion by 2025 [49]. Wearable ambulatory sensor devices can provide critical monitoring and diagnostic insights to prevent and treat health problems such as arrhythmias, coronary artery disease and diabetes [49,50]. The wearable devices can be used independently or in conjunction with other compact and portable equipment such as smartphones [51]. The wearables devices can be a convenient tool to diagnose asymptomatic or symptomatic AF [49]. A recent study showed the utility of these devices detecting AF. Among those device users who received notification of an irregular pulse, 34% had AF on subsequent electrocardiogram (ECG), and 84% of notifications were concordant with AF [52].

### 2.2. Mechanism of Detection in Heart Rhythm: Types of Sensors

It is important to understand the different technologies to achieve a greater understanding of both the clinical utility as well as the potential for error, limitations and challenges [48]. Accelerometer-based sensing systems collect data from all three axes of movements and infer important physiological parameters, including the heart rate (HR) using ballistocardiogram (BCG)-based approaches, which measure the repetitive movement of the human body caused by the heartbeat and the blood ejection. Additionally, phonocardiography or seismocardiography (SCG), which record cardiac sound and vibrations, respectively, can better detect cardiac structural abnormalities [53]. Furthermore, devices can measure both heart rate and rhythm via photoplethysmography (PPG), which measures changes in microvascular blood volume that translate into pulse waves and a tachogram recording by capturing the light intensity reflected from skin [54]. To refine precision, algorithms have been developed to differentiate AF from other benign irregular heart rhythms, such as bigeminy or premature beats, without requiring electrocardiographic corroboration [55]. However, there are mixed results in terms of accuracy when faced with different skin tones, skin moisture levels and the presence of tattoos [56,57]. Another way to measure heart rate and rhythm is by single-lead electrocardiogram (ECG), using different parts of a watch as both the positive and negative electrodes; its utility in analyzing more complex rhythms, however, is proven to be limited [49].

### 2.3. Types of Devices Available on the Market

There are various devices available for diagnosing arrhythmias, including AF (Figure 1). The conventional 12-lead ECG provides an accurate assessment of cardiac arrhythmias. Crucially, however, ECGs offer only a “snapshot” of electrical signals and can miss arrhythmias that occur outside this period. To overcome the problem, continuous ECG monitoring systems have been developed. Holter monitors are the mainstay device for the ambulatory detection of clinical arrhythmias. Nonetheless, these devices can offer only a limited duration of continuous monitoring [42]. The Zio Patch, for example, is a novel, single-lead ECG, lightweight, continuously recording ambulatory adhesive patch monitor suitable for detecting cardiac arrhythmias for up to two weeks [58]. Furthermore, the external loop recorders (ELRs) can be used in patients with a high probability of recurrent events of syncope [59]. A study by Locati et al. showed the diagnostic yield of external monitoring was 42.4% at one week, 57.2% at two weeks and 71.6% at four weeks [60]. In patients with unexplained syncope of suspected arrhythmic origin, if 4-week external ECG monitoring is not enough, a more expensive and minimally invasive implantable loop recorder (ILR) should be used. For patients with unexplained palpitations, the 4-week external ECG monitoring can provide a conclusive diagnosis in most cases. However, in a few instances, more prolonged monitoring by ILR would be required [60].

These devices, as mentioned above, are well-established tools for the diagnosis of arrhythmias, including atrial fibrillations. Although wearable devices are primarily worn for routine health monitoring, including heart rate, they can be used as an initial alarm to seek medical attention. These devices have the potential to identify the paroxysmal irregularities of heart rhythm even before the symptom onset. With the recent advancement of sensors used in these devices, the effectiveness has significantly increased. Depending on the type of sensors used, the wearable devices can be smart fabrics (e-textiles) or clothes, ECG sensors such as Apple Watches, near-field communication (NFC) sensors such as smartphones and photoplethysmography (PPG) sensors such as pulse oximeters [60]. In the last few years, technology has improved tremendously to simplify and popularize the use of these devices. Various straps, bands, watches and rings can be used for the purpose of continuous monitoring of heart rate, blood pressure and calorie expenditure daily [49]. We have witnessed the extensive use of consumer wearables during the COVID-19 pandemic for monitoring heart rate and oxygen saturation at home [61]. Undoubtedly, these devices have immense potential to become a reliable tool for monitoring and early diagnosis of various conditions, including AF. The application of artificial intelligence (AI) has further enhanced the efficacy of these devices [42].

Given the many potential benefits of utilizing wearable devices to detect and manage AF, it is no surprise that studies assessing their clinical utility abound. One study assessing the effect on detection of AF using a Zio Patch showed increased AF detection of 3.9% vs. 0.9% (absolute difference, 3.0% (95% CI, 1.8–4.1%)) over a four-month period [62]. Another study sought to validate deep neural network to detect AF using Apple Watch data with two cohorts: patients undergoing electrical or pharmacologic cardioversion and ambulatory patients with self-reported AF. Using ECG as the gold standard for diagnosis, researchers found a sensitivity of 98.0% and a specificity of 90.2% in the former subgroup, which is consistent with another study that assessed the performance of the Apple Watch for patients undergoing cardioversion and reported 93% sensitivity and 84% specificity [63]. However, in the latter group, which arguably is more representative of the ultimate application of this technology, results were more modest, with a sensitivity of 67.7% and specificity of 67.6% [64]. In addition, a large study that enrolled 419,297 participants to assess the ability of an Apple Watch to detect AF found a positive predictive value of 84%, but study sensitivity and specificity were not measured in the same study. Unfortunately, only 450 of 2161 notified of an irregular pulse returned ECG patches for more precise rhythm analysis, thus limiting generalizability [52]. Regarding a comparison between PPG-based technology versus ECG-based, there have been mixed results in terms of superiority [45,54]. Lastly, smartphone applications, although requiring more user participation and effort compared to passive detection, have been reported with sensitivity and specificity as high as 97.0% and 93.5%, respectively [65].

### 2.4. Advantages and Disadvantages of Wearable Devices

Wearable devices have several potential benefits to users. They are readily available; most come with a smartwatch or smartphone, which is already familiar to the individuals. They are convenient to use. Patients may choose to share the information with their providers, making it helpful for remote monitoring of the data without increasing the number of clinic visits [66]. Such convenience and remoteness of monitoring have become even more pivotal during the pandemic era. Continuous feedback from these devices may help individuals find motivation to maintain healthy lifestyles [49]. These devices can benefit patients with chronic conditions, including patients with cardiovascular risks, particularly elderly populations. As these devices can work as an initial screening, they have the potential to improve health care delivery and reduce costs significantly [66].

However, some questions need to be clarified regarding the use of these devices. The population that is at risk of having AF is mostly older people. We need to consider whether these people will find the devices easy to use. The accuracy of these devices needs to be monitored and titrated based on a standardized scale. After the initial diagnosis of AF by these devices, studies should be conducted to find out the next steps for the subsequent evaluation [55]. In addition, focusing on these devices for every alarm can result in unnecessary anxiety for the individuals using them. Some other barriers to more widespread utilization of ambulatory monitoring devices include issues related to reimbursement and insurance policies. Availability of health care facilities that can monitor the data from these devices and providers with training to obtain and interpret the data needs to be established [48].

### 2.5. The Prospective in the Future of Wearable Devices Development

With an aging population facing an already overburdened US healthcare system—not to mention the importance of virtual medicine in the era of the COVID-19 pandemic—the prevalence and development of more sophisticated wearable technologies that can detect AF appear inevitable [67]. However, as mentioned, there are serious concerns regarding the accuracy and reliability of these technologies despite persistent efforts to enhance and improve algorithms [68,69], making human oversight and clinical judgment a necessary component of the integration of these devices into our healthcare universe. Additionally, there are concerns regarding the lack of standardization of smartphones, which can also hinder reliability in the general population. Apart from these technical concerns, however, there are more general barriers to widespread implementation from the perspective of security, privacy and data ownership [69]. Ensuring patient trust and securing public willingness to use this technology are essential to actualize the potential for these developing innovative solutions. Lastly, as with all novel technologies, sincere effort must be made to prevent wearables from causing yet another health disparity to emerge between those with high and low socioeconomic status. Hopefully, with more data that favor widespread implementation of wearable technology, both in terms of cost and patient care, insurance companies will expand reimbursement so that benefits can be shared societally.

## 3. Artificial Intelligence and Machine Learning Utilizing Advanced Technology in View of Atrial Fibrillation Diagnosis

### 3.1. Overview of Algorithms

Artificial intelligence (AI) refers to systems or machines that mimic human intelligence to perform tasks and can iteratively improve themselves based on the information they collect. The term AI is often used interchangeably with its subfields, which include machine learning (ML). Machine learning is focused on building systems that learn or improve based on the data they consume. Categories of ML can be divided into supervised learning, unsupervised learning, semi-supervised learning, reinforcement learning and active learning tasks.

The application of artificial intelligence (AI) and machine learning (ML) in medicine has become important in intense exploration with increased cardiovascular disease, which is responsible for nearly a third of all deaths worldwide [70]. Cardiovascular diseases, such as atrial fibrillation (AF), affect up to 34 million people in the world [71], and patients with AF exhibit a higher risk of severe health consequences, including death and stroke. There has been considerable research in using machine learning (ML) to improve cardiovascular outcomes in patients. Researchers and clinicians can use AI/ML methods and datasets for diagnosis and disease classification, risk prediction and patient management. Therefore, earlier detection of AF with the use of anticoagulation therapy would mitigate the risk of stroke and other thromboembolic complications.

ML methods involve the scientific study of statistical models and algorithms that can progressively learn from datasets to perform results and achieve goals on a specific task. The processes of the ML pipeline consist of several steps, including data acquisition and preprocessing; feature extraction and selection; and selection of supervised or unsupervised learning methods. ML algorithms can be trained with a small input dataset, and then these trained algorithms can be applied to other large and variable datasets to predict specific cardiovascular diseases. With the introduction of better-trained ML algorithms, more accurate disease prediction would help clinicians obtain an improved diagnosis, classification and risk stratification.

### 3.2. Supervised Machine Learning for Atrial Fibrillation Detection

Supervised ML is training a model to relate input data and to learn a function that maps input data to target labeled outcomes of interest [72]. With a labeled training dataset, the problems can be further categorized into problems of regression and classification. Regression refers to predicting a continuous outcome when target variables are continuous real number values. Classification refers to predicting outcome labels on new data when the target variables are categorical variables. In supervised ML, training data consisting of input features and corresponding data labels (outputs) are provided to an ML algorithm. The ML algorithm then fits the ML model by learning the relationships between the features and the data labels, a process referred to as training. Once the model is trained, it will be able to make predictions from new data, a process referred to as testing. R-R intervals obtained through ECG recording can be translated into a Lorenz plot [73]. Subsequently, the data can be further presented to the ML algorithm [74].

The feasibility of signal-processed surface electrocardiography (spECG) with the basic use of traditional supervised ML as a diagnostic tool has been used to predict the presence of abnormal cardiac muscle relaxation [75]. An ML Cardiio Rhythm algorithm (supervised support vector machine) using facial and fingertip photoplethysmographic (PPG) data obtained from an iPhone camera was tested on 217 cardiology inpatients. The results show that the Cardiio Rhythm algorithm discriminated atrial fibrillation from sinus rhythm with 95% sensitivity, 96% specificity, PPV 92% and NPV 97%. Hence, it is feasible that detection of a facial PPG signal is able to determine pulse irregularity attributable to AF. The result of the study shows high sensitivity and specificity, while a low negative likelihood ratio for AF can be detected from facial PPG signals with the Cardiio Rhythm smartphone application [76]. The algorithm of the supervised random forest algorithm was trained using clinical variables from 481 CRT-P patients and tested on 595 CRT-D patients from the COMPANION trial [77]. The result shows that death or heart failure hospitalization was predicted within 12 months with an AUC of 0.74. Meanwhile, an eight-fold difference in all-cause mortality was shown between the top and bottom quartiles [78]. The application of supervised deep learning with convolutional neural networks (CNNs) was used to train with heart rate, activity level and ECGs from smartwatches of 7500 subjects, with testing on 24 patients. Compared with an insertable cardiac monitor, the smartwatch detected AF with episode sensitivity of 97.5%, PPV of 39.9% and duration sensitivity of 97.7% [79]. Analysis incorporates deep neural networks, which are a type of machine learning algorithm with multiple layers of processing that helps to yield higher accuracy in performing pattern recognition of data [63]. Supervised deep learning with recurrent neural network (RNN) combined with a supervised gradient boosting classifier was trained using 8183 single-lead ECGs and tested on 3658 subjects. The approach used classified ECG with an overall F1 score of 0.83, which has been validated against the 2017 PhysioNet/CinC Challenge dataset and ranked first in the PhysioNet/CinC competition [80]. In total, 30 days of cardiac implantable electronic device remote monitoring data in 3114 non-stroke controls and 71 stroke cases were compared against the past 30 days of remote monitoring before stroke. Three different types of supervised machine learning models were trained, including convolutional neural networks (CNNs), random forest and L1 regularized logistic regression (LASSO), in the study. The result shows that combining CHA2DS2-VASc with CNNs and random forests yielded a validation AUC of 0.696 and test AUC of 0.634, while CHA2DS2-VASc without CNNs and random forests only had an AUC of 0.5 or less in both datasets [81].

### 3.3. Unsupervised Machine Learning for Atrial Fibrillation Detection

Unsupervised ML models learn from clustering and association patterns of unlabeled input data without human intervention. Compared with supervised ML, unsupervised ML is the training of algorithms without definition of the output. Unsupervised ML does not train a model to predict labels from input data. Unsupervised ML instead quantifies natural patterns within input data, and it blinds to the labels of interest. Parsing out these patterns of unsupervised ML reveals an underlying structure to complex data which may help to identify relevant subgroups. One medical example of unsupervised ML was predicting or managing heart disease based on anomalies found within a battery of patient characteristics such as sex, age, body mass index (BMI), lab values and lifestyle, etc., of 9750 patients who participated with smartwatch PPG and tested on 51 patients undergoing cardioversion. The algorithm of the unsupervised approach was used in conjunction with deep learning neural network, followed by supervised classification (semi-supervised). The detection of AF was with an AUC of 0.97 (95% CI, 0.94–1.00; *p* < 0.001) in patients who underwent cardioversion and an AUC of 0.72 in self-reported ambulatory patients [63].

Ebrahimzadeh et al. performed a combination of feature extractions and a mixture of classifier approaches. Their model demonstrated 98.21% accuracy in predicting atrial fibrillation from a large database [82]. Machine learning allows large-scale screening and prediction of atrial fibrillation at the primary care setting level [83,84,85].

## 4. Prospective Future of Machine Learning for Atrial Fibrillation Diagnosis

Albeit in its infancy, ML has shown the potential to improve our ability to detect AF. Global efforts are on the march to further develop this technology. In the US, AI-enhanced electrocardiography predicted the future risk of AF compared to standard models [47]. In Germany, the eHealth-based Bavarian alternative detection of atrial fibrillation, or “eBRAVE-AF” trial, sought to find out the ability of AI to predict patients at risk for AF [86]. Technological giants (e.g., Apple, AliveCor, Fitbit and Honor Band 4) continue to push the boundaries of wearable technologies, improving signal and algorithm fidelity of their devices [87,88,89,90,91]. Despite significant advances, there continues to be substantial room for growth in this technology.

While wearable devices allow patients to incorporate preventive technology into their daily life, for their widespread adoption, issues of ease of use, effectiveness and socio-economic barriers remain. Efforts have focused on decreasing user input by developing textiles and implants capable of biophysical sensing [92]. Although the current performance of this technology does not allow it to replace the formal ECG in the diagnosis of AF, routine use of ambulatory monitoring will encourage patients to seek prompt medical care and ultimately reduce the burden on the health system. However, these devices tend to be cost-prohibitive and rely on wireless technology, which is not widely available even in developed countries [93]. Routine implementation permits scalability, lowering production costs and incentivizing the deployment of supporting technology.

Future improvement and miniaturization of biosensors will allow more data points from a single device. Current studies have focused on the isolated interpretation of physiologic parameters (ECG, imaging, clinical data), while traditional prediction models have shown remarkable predictive ability when incorporating different data sources [94,95,96,97]. Henceforth, integrating multiple parameters into a unified AI architecture will most likely improve the prediction accuracy and detection of atrial fibrillation. Furthermore, we can extrapolate these predictive models into different cardiac arrhythmias, such as ventricular tachycardias, which continue to be a substantial contributor to cardiac mortality [98]. The deep neural network currently used in many studies, with its incorporation of innumerable parameters, multi-layer data processes and self-learning capabilities, continues to demonstrate an advanced level of information analysis, pattern recognition and supremacy of accuracy. It will not be long until the sophisticated deep neural network is integrated with wearable devices to perfect accurate detection [99].

However, there are inherent limitations to our current method of machine learning. Neural networks operate as a “black box”, meaning they do not display the model’s path or “behavior”. In complex decision making, the “why” plays a more prominent role [99]. Denoting the reasoning allows clinicians to identify new diagnostic patterns, potentially furthering our understanding of human workings. These explanations are also a helpful tool in troubleshooting and refining existing AI.

Machine learning might expand its horizon from predictive models. It might one day be the key to selecting novel therapeutic strategies and the key to uncovering new mechanisms of disease. There is untapped potential in this field that remains in its early stages. As we continue to explore the interlaced workings of machines and clinical data, we might one day have the potential to mount therapeutic strategies before clinical presentation [100].

## 5. Conclusions

The development of wearable devices and artificial intelligence allows higher detection and faster diagnosis of atrial fibrillation, especially in an asymptomatic patient population. Future large-sized randomized clinical studies are needed to determine the most optimal strategies for diagnosis of atrial fibrillation.

## Figures and Tables

**Figure 1 diagnostics-12-00689-f001:**
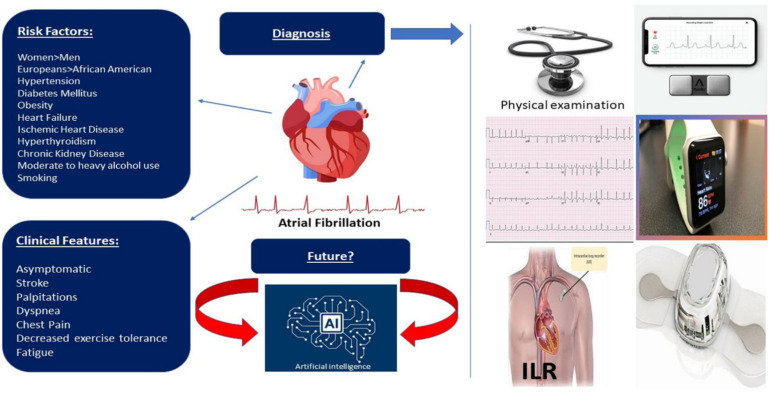
Wearable devices incorporated into diagnostic plan to help pick up silent and paroxysmal atrial fibrillation. Development of artificial intelligence and machine learning will play a major role in the process of diagnosis.

**Table 1 diagnostics-12-00689-t001:** Examples of wearable devices for the detection of arrhythmias available on the market.

Device Name	Device Type	Use	Key Points
Zio Patch	Patch	Lasts 14 days at a time without needing to change battery; then sent in for analysis and interpretation	Minimally intrusive to daily activities, water-resistant, hygienic (single use only). High cumulative consumer costs (due to non-reusability), no real-time analysis or transmission
Nuvant MCT	Patch	Lasts 7.5 days but can have unit replaced for 30-day total duration	Real-time analysis and transmission of ECG data, but real-time information not available to user
BodyGuardian	Patch	Continuously records, stores and can periodically transmit the clinical data for up to 30 days at a time	Small, discreet, wireless monitor. Attaches to the chest via the disposable strip. Real-time tracking but analysis result will be known to the user within a week or two after the test
BardyDx CAM	Patch	Wire-free monitoring device that continuously records heartbeat	Optimizes P-wave signal capture, results in improved ECG resolution, provides more information about heart rhythm, leads to more clinically relevant diagnoses
BioTel Heart	Patch	Gathers cardiac rhythm data from the sensor via Bluetooth, then sends these ECG data via a wireless connection	Supports continuous patient oversight, helps early detection of potential adverse events
MediBioSense MBD HealthStream	Patch	MediBioSense is a real-time cardiac rhythm monitor. ECG reporting utilizing VitalPatch wearable sensor provides real-time 24/7 full medical analytics	Includes GPS tracking, voice calls to health centers, fall detection, SOS button and heart rate monitoring
Kardia Mobile	Patch	Portable sensor works with most smartphones and tablets. Captures cardiac arrhythmia in real time. FDA-approved for detection of atrial fibrillation	Rhythm strip will be analyzed for atrial fibrillation, bradycardia, tachycardia, premature ventricular complexes, sinus rhythm with wide QRS and sinus rhythm with supraventricular ectopy
Apple Watch	Wristwatch	Active whenever Watch is in use; lightweight; user-friendly	High specificity in identifying patients with silent AF
SmartCardia INYU	Wristwatch	Lasts 14 days; real-time transmission and analysis	Reusable and commercially cost-friendly
PulseSmart	Smartphone camera-based app	Pulse waveform analysis from finger on smartphone camera and flashlight	Sensitivity, specificity and accuracy all >93%; discrete, non-continuous monitoring
ECG Check	Smartphone case	Consists of 2 metal electrodes, connected to a smartphone, that one places a single finger on for 30 s	Users are told whether the heart rhythm is normal or abnormal and can choose to send their ECG to a medical professional for further interpretation and management; limitation is that this is non-continuous monitoring

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
