# Peer review of "Current Advancement in Diagnosing Atrial Fibrillation by Utilizing Wearable Devices and Artificial Intelligence: A Review Study"

_diagnostics, 2022, doi:10.3390/diagnostics12030689_

Round 1
Reviewer 1 Report
In this review, Dr. Wang and colleagues discussed about the emergence of ECG wearables, AI and their relevance for an early diagnosis of AF. Overall, this is a nicely written manuscript. I do have some minor suggestions:
- In Section 1.1, while the authors nicely explained about paroxysmal and persistent AF, the authors did not explain about long-standing persistent and permanent AF. Please add them.
- "treatment of AF can be categorized into three major categories: rate control, rhythm control and anticoagulation based on the CHA2DS2-VASc score for stroke prevention." So, HAS-BLED score does not have any importance/relevance anymore? Please clarify.
- "...pharmacological cardioversion with class 3B anti-arrhythmic drugs" which AAD classification is used here? I don't think Vaughan-William has 3B? Please clarify.
- I think this review correlates very well with this recent review by Heijman et al. (PMID: 33890620), especially the ML part. Consider incorporating some information from that review as well.
- In Page 5, AliveCor KardiaMobile was classified as a smartphone case. While this might be true for the first generation, it is not valid anymore since they also have the 6-lead version, which is not possible to be placed at the back of a smartphone. Please revise.
- As far as I understand, the use of AI is currently in the data post-processing only (e.g., in subsequent studies using those devices). For example, raw data from Apple Watch can be extracted and AI is used to analyze the rhythm. Am I correct? Is there any market-ready wearable shown in Table 1 that already integrates AI? It would be insightful to add this information in that table or somewhere else in the text.
- Nowadays, it becomes more often to separate deep learning / neural network from unsupervised learning. I think the authors should also do this and distinguish those two. Also, please provide more examples because I am quite sure that neural network would be the AI-of-choice when it comes to the integration with wearables.
Author Response
Thank you very much for the detailed review and recommendation. We really appreciate the input and made the adjustment as in the updated manuscript.
- In Section 1.1, while the authors nicely explained about paroxysmal and persistent AF, the authors did not explain about long-standing persistent and permanent AF. Please add them. Both definitions were added, currently modified with comprehensive Afib classification.
- "treatment of AF can be categorized into three major categories: rate control, rhythm control and anticoagulation based on the CHA2DS2-VASc score for stroke prevention." So, HAS-BLED score does not have any importance/relevance anymore? Please clarify. Both scores are important in decision-making for starting or discontinuing anticoagulation nowadays. Thank you for pointing out the missing important relevant score here. The change was made in the manuscript. Please see the attachment.
- "...pharmacological cardioversion with class 3B anti-arrhythmic drugs" which AAD classification is used here? I don't think Vaughan-William has 3B? Please clarify. This was indicating Vaughan-Williams anti-arrhythmic drug classification 3. The change was made in the manuscript. Please see the attachment.
- I think this review correlates very well with this recent review by Heijman et al. (PMID: 33890620), especially the ML part. Consider incorporating some information from that review as well. Will incorporate this article into the manuscript. Please see the attachment.
- In Page 5, AliveCor KardiaMobile was classified as a smartphone case. While this might be true for the first generation, it is not valid anymore since they also have the 6-lead version, which is not possible to be placed at the back of a smartphone. Please revise. Thank you for pointing out this information. Will update the categorization in the table. Please see the attachment.
- As far as I understand, the use of AI is currently in the data post-processing only (e.g., in subsequent studies using those devices). For example, raw data from Apple Watch can be extracted and AI is used to analyze the rhythm. Am I correct? Is there any market-ready wearable shown in Table 1 that already integrates AI? It would be insightful to add this information in that table or somewhere else in the text. Most wearable devices are now utilizing machine learning and deep neural network, which is part of artificial intelligence, to detect and analyze AF. However, more fields can still be involved in future device development.
- Nowadays, it becomes more often to separate deep learning / neural network from unsupervised learning. I think the authors should also do this and distinguish those two. Also, please provide more examples because I am quite sure that neural network would be the AI-of-choice when it comes to the integration with wearables. Modifications made to separate deep neural network and unsupervised learning. Comments also made for the utility of neural network.

Reviewer 2 Report
This is a well-written manuscript and addresses the role of AI in analyzing wearable device-detectable arrhythmia.
Few comments:
- can they separate those cases of AF who were no symptoms (sub-clinical) vs clinically significant AF
- How to manage the sub-clinical AF?
- How close are these devices to the clinical application?
- Please describe the characteristics of wearable devices and how to differentiate AF and sinus rhythm?
- Was there any association with stroke?
Author Response
Thank you so much for the insightful review and recommendation. We adjusted the updated manuscript. We sincerely appreciated your time.
- can they separate those cases of AF who were no symptoms (sub-clinical) vs clinically significant AF? The current wearable devices cannot separate subclinical or clinical AF since those are programmed to detect the AF rhythm if present. The significance of wearable devices to detect AF is to be able to detect subclinical AF, and monitor AF for clinical AF.
- How to manage the sub-clinical AF? Our aim is to improve detection rate of AF, including sub-clinical AF. The management of sub-clinical AF is not discussed in this paper.
- How close are these devices to the clinical application? The hope is that through improving AI/machine learning data analysis, Afib detection rate can be improved to the same level of sensitivity and specificity of the goal standard AF detection using ECG.
- Please describe the characteristics of wearable devices and how to differentiate AF and sinus rhythm? Different characteristics of wearable devices are listed in table 1 of our paper. We have also described the different type of sensors in various wearable devices for AF monitoring and detection.
- Was there any association with stroke? The aim of our paper is to summarize the current advancement in diagnosis AF using wearable devices. The association between increased detection of AF by wearable devices and stroke risk reduction will need to be evaluated in future studies.

Reviewer 3 Report
Very interesting subject and very actual, with practical implications.
I noticed some medical inaccuracies and I mentioned them directly into the text
Also, my comments are inserted into the text

Author Response
Thank you so much for giving us an insightful review and recommendation. We updated the manuscript according to your suggestion. Please see the updated manuscript as attached. Thank you again for your time.
